# Expression of CTLA-4 and CD86 Antigens and Epstein-Barr Virus Reactivation in Chronic Lymphocytic Leukemia—Any Link with Known Prognostic Factors?

**DOI:** 10.3390/cancers14030672

**Published:** 2022-01-28

**Authors:** Ewelina Grywalska, Michał Mielnik, Martyna Podgajna, Anna Hymos, Jarosław Ludian, Agnieszka Rolińska, Krzysztof Gosik, Wojciech Kwaśniewski, Barbara Sosnowska-Pasiarska, Jolanta Smok-Kalwat, Marcin Pasiarski, Agnieszka Stelmach-Gołdyś, Stanisław Góźdź, Jacek Roliński

**Affiliations:** 1Department of Experimental Immunology, Medical University of Lublin, 20-093 Lublin, Poland; ewelina.grywalska@umlub.pl (E.G.); 50618@umlub.pl (M.P.); anna.hymos@umlub.pl (A.H.); jaroslaw.ludian@umlub.pl (J.L.); krzysztof.gosik@umlub.pl (K.G.); 2Department of Hematooncology and Bone Marrow Transplantation, Medical University of Lublin, 20-081 Lublin, Poland; 3Department of Applied Psychology, Medical University of Lublin, 20-093 Lublin, Poland; agnieszka.rolinska@umlub.pl; 4Department of Gynecologic Oncology and Gynecology, Medical University of Lublin, 20-081 Lublin, Poland; wojciech.kwasniewski@umlub.pl; 5Department of Oncocardiology, Holy Cross Cancer Centre, 25-734 Kielce, Poland; barbara.pasiarska@onkol.kielce.pl; 6Department of Clinical Oncology, Holy Cross Cancer Centre, 25-734 Kielce, Poland; jolantasm@onkol.kielce.pl (J.S.-K.); stanislawgozdz1@gmail.com (S.G.); 7Department of Immunology, Faculty of Health Sciences, Jan Kochanowski University, 25-317 Kielce, Poland; marcinpasiarski@gmail.com (M.P.); astelmach3@o2.pl (A.S.-G.); 8Department of Hematology, Holy Cross Cancer Centre, 25-734 Kielce, Poland; 9Faculty of Medicine and Health Sciences, The Jan Kochanowski University, 25-516 Kielce, Poland; 10Department of Clinical Immunology, Medical University of Lublin, 20-093 Lublin, Poland; jacek.rolinski@umlub.pl

**Keywords:** Epstein-Barr virus, chronic lymphocytic leukemia, CTLA-4, CD86, prognostic factors

## Abstract

**Simple Summary:**

Chronic lymphocytic leukemia (CLL) accounts for one-third of all leukemias. The Epstein-Barr virus (EBV) has the ability to transform B-cells into cancer cells. A history of EBV infection increases the chances of acquiring CLL and it worsens the prognosis in CLL. We tried to assess whether EBV affects the course of CLL by deregulating the CTLA-4/CD86 pathway. The expression of CTLA-4 and CD86 on immune cells in patients with CLL has been evaluated and linked to indicators of EBV infection and clinical outcomes. Our studies have shown that anergy, which is expressed by inhibition through the interaction of CTLA-4 and CD86, is an important mechanism leading to the inhibition of the antitumor response and CLL progression.

**Abstract:**

Infection with Epstein-Barr virus (EBV) worsens the prognosis in chronic lymphocytic leukemia (CLL), but the underlying mechanisms are not yet established. We intended to assess whether EBV affects the course of CLL by the deregulation of the CTLA-4/CD86 signaling pathway. We used polymerase chain reaction to measure the load of EBV DNA in the blood of 110 newly diagnosed patients with CLL. The expression of CTLA-4 and CD86 antigen on lymphocytes was assessed with flow cytometry. Additionally, CTLA-4 and CD86 serum concentrations were measured through enzyme-linked immunosorbent assays. Fifty-four percent of the patients had detectable EBV DNA [EBV(+)]. In EBV(+) patients the CTLA-4 and CD86 serum concentrations and their expressions on investigated cell populations were significantly higher than in EBV(−) patients. EBV load correlated positively with unfavorable prognostic markers of CLL and the expression of CTLA-4 on CD3+ lymphocytes (r = 0.5339; *p* = 0.027) and CD86 on CD19+ cells (r = 0.6950; *p* < 0.001). During a median follow-up period of 32 months EBV(+) patients were more likely to require treatment or have lymphocyte doubling (*p* < 0.001). Among EBV(+) but not EBV(−) patients, increased expressions of CTLA-4 lymphocytes were associated with elevated risks of progression. We propose that EBV coinfection may worsen prognosis in CLL patients, partly due to EBV-induced up-regulation of CTLA-4 expression.

## 1. Introduction

Chronic lymphocytic leukemia (CLL) accounts for approximately one-third of all leukemias in the western hemisphere and is recognized as being the most common leukemia [1]. CLL is characterized by an unconstrained proliferation of morphologically mature, albeit functionally impaired B-cells in the peripheral blood, lymph nodes, bone marrow, liver, and spleen, leading to lymphocytosis, lymphadenopathy, cytopenia, or organomegaly [2]. Every third CLL patient will never require any treatment and can have a very long survival. The other patients, however, succumb to aggressive leukemia after just several months [3]. The course of CLL depends on the clinical stage and some molecular factors, such as CD38 or ZAP70 expression on leukemia cells or deletion 17p leukemia [4]. Certainly, other factors are also involved.

The Epstein-Barr virus (EBV) infects over 90% of people worldwide. It has the capability to transform B-cells into neoplastic ones [5,6]. A history of EBV infection (mononucleosis) increases the likelihood of developing CLL [7]. Patients with CLL and EBV coinfection are more likely to progress into diffuse large B-cell lymphoma (Richter’s transformation) [8]. Moreover, the role of EBV was studied in NK-type lymphoproliferative diseases [9,10]. Previously our team demonstrated that the EBV DNA presence in the blood of patients with CLL was associated with a more serious clinical burden of the disease and a shorter time to treatment initiation [11]. Our team also suggested that EBV-induced up-regulation of PD-1-PD-L1 expression can be responsible for worsening the prognosis in patients with CLL [12]. We believe there are many more mechanisms involved that need further exploration.

It has been proven that chronic viral infections lead to T-cell exhaustion impairing the immune response towards cancer cells. It is very likely that EBV infection may have similar effects [13].

The CTLA-4 (CD152) molecule belongs to the family of type I membrane receptors, to the immunoglobulin superfamily. It is made of 186 amino acid chains: 37 amino acid residues make up the cytoplasmic part, 24 constitute a fragment anchored in the cell membrane, and 125 constitute the extracellular domain [14]. The expression of the CTLA-4 molecule, which inhibits the activation process, in CD4+ and CD8+ T cells is mediated by antigenic stimulation of the TCR/CD3 receptor [15]. Its insignificant amounts can be found on the surface of resting T lymphocytes. Moreover, CD19+ B lymphocytes subjected to antigenic stimulation synthesize this molecule. The CTLA-4 molecule is expressed on T lymphocytes that have the CD28 antigen, but its affinity for the ligand B.7-1 (CD80) and B.7-2 (CD86) is 10–50 times greater [16,17]. CTLA-4 ligation, resulting in the lymphocyte anergy state, leads to a decrease in the synthesis of IL-2, IL-3, GM-CSF, and IFN-gamma, and also increases the production of TGF-beta and does not significantly affect the concentration of IL-4 [18]. Already within the first hour of stimulation, the mRNA synthesis for CTLA-4 increases, while the peak expression is found after 48–72 h [19]. CTLA-4 promotes the arrest of lymphocytes in the G0/G1 phase of the cell cycle, inhibiting the synthesis of cyclin D3 and cdk4/cdk6 kinases and the degradation of the inhibitory p27 protein, as well as enhancing the expression of cyclin D2 [20,21,22,23]. A lymphocyte that enters an anergy state is not activated after recognition of a specific antigen, even if it receives sufficient co-stimulatory signals to activate a virgin lymphocyte.

CD86 (B70/B7.2) has been identified as the second—besides CD80 (B7/B7.1)—ligand for CD28 and CTLA-4 antigens, playing an important role in T cell costimulation and primary immune response. It belongs to the immunoglobulin superfamily and is expressed on monocytes, dendritic cells, activated T, B, and NK lymphocytes. The CD86 antigen-coding region contains several genes involved in tumorigenesis [24,25].

CTLA-4 expression has been found on the surface of tumor-infiltrating macrophages and in neoplastic tissues, where the molecule contributes to the suppression of the specific anti-tumor immune response. Hock et al. confirmed the participation of the soluble form of the ligand for CTLA-4—CD80 (soluble CD80, sCD80) in the pathogenesis of hematological neoplasms [26]. Scrivener et al. reported a reduced number of T cells expressing CTLA-4, CD25, and CD28 molecules [27]. Disturbed expression of CD28 and CTLA-4 molecules was found on T lymphocytes in CLL, which may be responsible for the deficiency of cellular immunity in this disease [16]. The frequency of the CTLA-4 G allele was highest in CLL patients who developed autoimmune hemolytic anemia [28]. Kosmaczewska et al. were the first to conduct studies on the surface and cytoplasmic expression of CTLA-4 in CD19+/CD5+ CLL B cells in relation to clinical parameters and polymorphism of genes encoding proteins regulating the cell cycle, showing the relationship of this molecule with the inhibition of the CLL cell cycle [29].

Both PD-1 and CTLA-4 expression are elevated on the surface of hepatitis C virus (HCV) antigen-specific CD4+ T cells [30]. Anti-CTLA-4 monoclonal antibody (tremelimumab and ipilimumab) has been found to block the interaction of CTLA-4 with CD80 and CD86, which may be used in the treatment of impaired cellular immunity in the course of viral infections and neoplastic diseases, e.g., melanoma [31,32]. Regulatory T cell depletion and CTLA-4 blockade enhance cytotoxicity against autologous EBV-transformed cell lines [32]. Parks et al. have shown that in patients with systemic lupus erythematosus, the response to reactivation of EBV infection is manifested by elevated levels of anti-EBV antibodies in the IgA and IgG classes and depends on the CTLA-4 genotype [33]. There are no reports in the available literature on the role of CTLA-4 and CD86 in the response to EBV in patients with CLL.

The goal of our work was to assess if EBV can worsen the course of CLL by deregulating the CTLA-4/CD86 pathway. In order to achieve this, we gauged the expression of CTLA-4 and CD86 on immune cells in patients with CLL, and we linked the expression of these protein indicators of EBV infection and clinical outcomes.

## 2. Materials and Methods

### 2.1. Patients

We included 110 newly diagnosed, treatment-naïve, patients with CLL (48 women and 62 men) and 20 healthy sex and age-matched volunteers (9 women and 11 men) for the control group. The National Cancer Institute criteria were used to diagnose CLL [34]. We excluded patients receiving immunomodulatory treatment or blood transfusions, with any signs of infection ≤2 months before enrolment, or diagnosed with allergic or autoimmune diseases. The median follow-up was 32 months (range: 10.5–74 months). The ethics committee of the Medical University of Lublin approved the study (KE-0254/227/2010, 6 December 2010). We obtained written informed consent from all participants. This study was performed strictly adhering to the principles of the Declaration of Helsinki.

### 2.2. Preparation of Material

Fifteen milliliters of peripheral blood in EDTA-coated tubes and an additional 10 mL in tubes with a clot activator (Sarstedt, Nümbrecht, Germany) were collected. Peripheral blood mononuclear cells (PBMCs) were isolated with the use of density gradient centrifugation from EDTA-coated tubes. Five milliliters of whole blood diluted with 5 mL of normal saline on 5 mL of Ficoll-Paque™ (Milteny Biotec, Bergisch-Gladbach, Germany) was layered in 15 mm tubes and centrifuged at 400× *g* for 30 min. With the use of a Pasteur pipette, PBMCs were removed from buffy coats. Trypan blue (0.4% trypan blue solution; Sigma-Aldrich, Hamburg, Germany) was utilized to count PBMCs and assay them for viability. Only PBMCs with a viability ≥95% were deemed eligible. Serum from the tubes with a clot activator was frozen and stored at −80 °C until used.

### 2.3. Immunophenotyping

The whole blood after erythrocyte lysis was subjected to immunophenotyping. Blood samples were incubated in the dark at room temperature for 20 min with monoclonal antibodies labeled with fluorescein isothiocyanate (FITC), phycoerythrin (PE), or CyChrome (PE-Cy5). The blood cells were stained using antibodies (BD Biosciences, San Jose, CA, USA) against the following antigens: CD86 (PE), CTLA-4 (PE-Cy5—eBioscience, Waltham, MA, USA), CD3 (FITC, PE, PE-Cy5), CD4 (FITC, PE-Cy5), CD8 (FITC, PE-Cy5), and CD19 (FITC, PE, PE-Cy5). We stained CD19+ cells with antibodies against CD38 and ZAP-70 as described by Hus et al. [35]. The cut-off for ZAP-70 positivity was ≥20%, and the cut-off for CD38 positivity was ≥30%. The cells were washed and analyzed by flow cytometry (BD FACSCalibur, San Jose, CA, USA) after incubation. Twenty thousand events were acquired and analyzed with the CellQuest Pro software for all measurements. We verified staining specificity and separated cell populations with the use of isotype-matched antibodies. We analyzed the percentages of different surface markers expression. The flow cytometric analysis of the expression of CD86 and CTLA-4 is illustrated in Appendix A.

### 2.4. Leukemia Cell Genotyping

Leukemia interphase fluorescence in situ hybridization (I-FISH) was utilized in order to genotype leukemia cells. PBMCs were cultured for 24 h in RPMI 1640 without stimulation with mitogen. Then, they were subjected to hypotonic treatment and fixated with methanol:acetic acid (3:1). Followingly, cell suspensions were placed onto microscope slides and used directly for I-FISH. Commercially available probes were used: Vysis LSI ATM SpectrumOrange, CEP 11 SpectrumGreen Probe, LSI TP53 SpectrumOrange, and CEP 17 SpectrumGreen (Abbott Molecular Europe, Wiesbaden, Germany). At least 200 nuclei per probe were analyzed. A cut-off of 2.5% (mean ± SD) was set for positive values for normal controls.

### 2.5. DNA Isolation and Calculation of EBV Load

Following the manufacturer’s instructions, we isolated DNA from 5 × 10^6^ PBMCs with the QIAamp DNA Blood Mini Kit (QIAGEN, Hilden, Germany). The BioSpec-nano spectrophotometer (Shimadzu, Kyoto, Japan) allowed us to verify the concentration and purity of the isolated DNA. Then, we calculated the number of EBV DNA copies in PBMCs with the ISEX variant of the EBV polymerase chain reaction (PCR) kit (GeneProof, Brno, Czech Republic). Real-time PCR (RT-PCR) was used to analyze EBV DNA qualitatively and quantitatively. A specific conservative DNA sequence for the *EBV nuclear antigen 1* (*EBNA-1*) gene was amplified with PCR. The number of viral DNA copies per μL of eluent was expressed as the viral DNA copy number per μg of DNA after being adjusted for the efficiency of DNA isolation. Duplicate examination of all samples was performed. A negative control was created with the use of a sample of pure buffer used for DNA elution in every case. All samples below 10 EBV DNA copies per μL were considered EBV-negative [EBV(−)] in accordance with the used detection method. The 7300 Real-Time PCR System (Applied Biosystems, Foster City, CA, USA) was used for PCR testing. The reaction was conducted on MicroAmp^®^ Optical 96-Well Reaction Plates (Life Technologies, Carlsbad, CA, USA) with MicroAmp^®^ Optical Adhesive Film (Life Technologies, Carlsbad, CA, USA).

### 2.6. Measurement of EBV-Specific Antibodies in Serum

Using the commercial enzyme-linked immunosorbent assays (ELISA; IBL International, Hamburg, Germany), we detected EBV-specific antibodies. We analyzed IgA, IgM, and IgG antibody classes against the early antigen (EA), viral capsid antigen (VCA), and EBNA-1. ELISA plates were read with the ELISA Reader Victor TM3 (PerkinElmer, Waltham, MA, USA). The manufacturer-specified cut-offs were adopted.

### 2.7. Measurement of Lactic Dehydrogenase and Beta-2 Microglobulin Concentrations in Serum

We measured the concentrations of lactic dehydrogenase (LDH) and beta-2 microglobulin (B2M) in serum using clinical-grade assays in local laboratories.

### 2.8. Statistical Analysis

Normality was tested with the Shapiro–Wilk test. Analysis of variance (ANOVA) or the Kruskal–Wallis test was used to compare variables, as appropriate. Post hoc comparisons were performed with the Dunn’s test, corrected for false discovery rate with the Benjamini–Hochberg method. Fisher’s exact test was used to compare categorical variables. The assessment of correlations between pairs of variables was performed with Spearman’s rank correlation coefficient (rho). The time to treatment initiation and the time to lymphocyte doubling between EBV(+) and EBV(−) patients and between patients with a high and low expression of CD86 and CTLA-4 within the subgroups of EBV(+) and EBV(−) patients were compared with the application of survival analysis. The classification of CD86 and CTLA-4 expression as high or low was based on medians specific for the subgroups of EBV(+) and EBV(−) patients. The Kaplan–Meier curves comparison was supported by the log-rank test. *p* < 0.05 was considered significant. The R software (R Core Team and the R Foundation for Statistical Computing, Vienna, Austria) version 3.0.2 (https://www.r-project.org, accessed on 2 December 2021.) was used to complete all calculations.

## 3. Results

EBV(+) patients with CLL presented more advanced disease, having a greater clinical disease burden (RAI and BINET scores) and higher values of unfavorable prognostic factors (B2M, LDH, percentage of CD19+ CD38+ and CD19+ ZAP70+ cells, deletion 11q) compared to EBV(−) patients. The white blood cell and lymphocyte counts were comparable in both EBV(+) and EBV(−) patients with CLL (Table 1).

The serum concentrations and CTLA-4 and CD86 expressions on investigated cell populations were significantly higher in EBV(+) and EBV(−) patients than in subjects from the control group. Moreover, this difference was also pronounced when comparing EBV(+) patients to EBV(−) patients (*p* < 0.05 for all comparisons, Table 2 and Table 3).

In patients with CLL and EBV coinfection, EBV load correlated positively with LDH (r = 0.3743; *p* = 0.010; Figure 1A, B2M (r = 0.3954; *p* = 0.005; Figure 1B), and the percentages of CD19+ ZAP70+ cells (r = 0.3211; *p* = 0.020; Figure 1C).

EBV load correlated positively with unfavorable prognostic markers of CLL and the expression of CTLA-4 on CD3+ lymphocytes (r = 0.5339; *p* = 0.027, Figure 2A) and CD86 on CD19+ cells (r = 0.6950; *p* < 0.001, Figure 2B).

During a median follow-up period of 32 months, EBV(+) patients were more likely to require to initiate the treatment or have the lymphocyte doubling (*p* < 0.001). The increased expressions of CTLA-4 on CD4+ and CD8+ cells were associated with elevated risks of progression, treatment initiation. and lymphocyte doubling among EBV(+), but not EBV(−), patients (*p* ≤ 0.020).

The percentage of CD4+/CTLA-4+ lymphocytes ≤ 12.62% was a factor that significantly increased the probability of progression-free survival after the diagnosis of CLL (*p* = 0.008; Figure 3A). Moreover, the percentage of CD8+/CTLA-4+ lymphocytes ≤ 14.45% was associated with an increased probability of progression-free survival after the initiation of treatment (*p* = 0.006; Figure 3B). The percentage of CD3+/CTLA-4+ lymphocytes ≤ 5.02% turned out to be a factor increasing the probability of progression-free survival after the initiation of treatment (*p* = 0.007; Figure 3C).

The percentage of CD4+/CTLA-4+ lymphocytes ≤ 12.62% was also a factor that significantly increased the probability of survival without doubling of the lymphocytosis (Figure 4A). Moreover, the percentage of CD8+/CTLA-4+ lymphocytes ≤ 14.45% was associated with an increased probability of survival without doubling of the lymphocytosis (Figure 4B).

## 4. Discussion

Slightly more than half of the patients in our study had detectable EBV DNA at baseline, and these patients presented with more severe disease than the remaining patients, based on both clinical status (RAI and Binet scores) and laboratory markers of disease burden (LDH, B2M, ZAP70, CD38). These early differences resulted in worse longitudinal outcomes in EBV(+) patients with CLL, which supports our previous findings [11].

This study upheld the association between EBV infection and worse outcomes in patients with CLL. Interestingly, this instance may be caused by an EBV-induced upregulation of CTLA-4 and CD86 expression on host immune cells. Within the group of CLL subjects, those with detectable EBV DNA had increased CTLA-4 and CD86 expression.

There are few reports on the expression of the CTLA-4 antigen on the surface of T cells in CLL in the available literature. Frydecka et al. showed that significant amounts of these molecules were present in patients on CD4+ and CD8+ T cells [16]. Our results support previous findings as they also showed significantly higher percentages and absolute numbers of CD4+/CTLA-4+ and CD8+/CTLA-4+ T lymphocytes in patients with CLL than in the control group. It seems that the high expression of the CTLA-4 molecule in both Hodgkin’s lymphoma and CLL significantly influences the biology and mechanisms of cellular immune disorders in these diseases [36]. Our research showed that a low percentage of CD3+/CTLA-4+ T cells (both CD4+/CTLA+ T cells and CD8+/CTLA-4+ T cells) is more common in people with a longer period of progression-free survival, as well as with a longer doubling time of lymphocytosis, and therefore, a favorable prognostic factor. Moreover, a positive correlation was found between the number of EBV DNA copies and the number of CD3+/CTLA-4+ cells, which may suggest that the cause of increased CTLA-4 expression is EBV infection. In our research carried out in vitro, it was found that after 72 h-long stimulation, the expression of that molecule increases on T lymphocytes in healthy people, while in patients with CLL this effect is achieved after 24 h and lasts longer—up to 120 h compared to 96 h in the control group [16]. Our study also showed that the maximum increase in the number of CD3+ T cells expressing CTLA-4 was achieved in patients with CLL without molecular markers for virus presence after 72 h of stimulation with EBV peptides. In patients with EBV(+) CLL, such changes were not observed; the significant increase in CTLA-4 expression was only achieved by culturing PBMCs in the presence of PMA and ionomycin. Most likely, the addition of EBV antigens, which are constantly present in patients from this group, was not able to induce an additional increase in CTLA-4 synthesis. As a result of stimulation of the samples with EBV peptides and PMA with ionomycin, no significant differences were found in the percentage of CD3+/CTLA-4+ lymphocytes in the EBV(−) patients and the control group. Therefore, it seems that in individuals who are not exposed to viral replication and protein synthesis, EBV antigens stimulate cells with a similar power to standard use mitogens.

CTLA-4 expression is induced during the activation of T and B lymphocytes and may also be present on the surface of neoplastic cells, which then indicates partial activation and may be directly related to the pathogenesis of the disease [37]. Reports by various authors have shown that CLL B lymphocytes have signs of activation, which is also manifested by high CTLA-4 expression on the surface of these cells compared to normal B lymphocytes [29,38]. These studies also showed a positive correlation between the percentage of CD5+/CD19+ CTLA-4+ leukemic B cells and the number of these cells expressing cyclin D2, induced during the early G1 phase of the cell cycle [29]. However, lymphocytes did not enter the S phase due to the simultaneous inhibition of cyclin D3 synthesis and the arrest of degradation of the inhibitory p27 protein by CTLA-4 [21]. The negative correlation between the percentage of CD19+/CD5+/CTLA-4+ B cells and the percentage of CD19+/CD5+/cyclin D3+ B cells and the positive correlation between the percentage of CTLA-4+ and p27^KIP1^ expressing leukemic cells is consistent with this observation [29]. Moreover, Suwalska et al. showed that the polymorphism of some genes for CTLA-4, ICOS, and CD28 is associated with the development of CLL [39]. On the other hand, according to Mittal et al., CLL B cells with reduced CTLA-4 expression were characterized by a greater ability to proliferate and survive, a higher expression of STAT1, NFATC2, c-Myc, Ki-67, Bcl-2 molecules, as well as the phosphorylation of STAT1 and c-Fos [40]. These dependencies were closely related to the microenvironment from which CLL B lymphocytes were derived or in which they were cultured [40]. An older study of this syndrome showed that in people with the CD38+ phenotype who had lower CTLA-4 expression, leukemia had a mild course [41]. Our studies did not show any relationship between the expression of the CTLA-4 antigen on CD19+ B lymphocytes and the clinical course of the disease, as opposed to the expression of this molecule on T lymphocytes.

The evaluation of the ligand for CTLA-4—CD86 molecules showed that in patients with CLL EBV(+) the percentage of CD19+/CD86+ lymphocytes significantly exceeded the values shown in EBV(−) patients and healthy people. Additionally, our research showed that in EBV(+) patients, the percentage of CD19+/CD86+ lymphocytes strongly positively correlated with the concentration of anti-EBV EBNA-1 antibodies in the serum IgA class and with the number of EBV DNA copies in 1 µg of DNA isolated from PBMCs. These relationships were one of the most pronounced in the presented work. Grandjenette et al. did not show increased expression of CD80 and CD86 antigens on CLL leukemic lymphocytes [42]. Stimulation of the TLR7 receptor in leukemic lymphocytes causes them to obtain a phenotype of antigen-presenting cells with increased expression of CD80 and CD86, and in normal cells, it causes an immune response through the synthesis of type I interferons [43,44,45]. Valente et al. showed that TLR7 activation enhances the expression of Epstein-Barr virus LMP-1 protein, and IFN regulatory factor (IRF) 7 is involved in the stimulation process [46]. However, TLR7 activation does not induce the production of interferon by EBV-infected lymphocytes but enhances the action of TLR3 and TLR9 in this process [46]. EBV also uses TLR-7 transmission to enhance B cell proliferation and modulate IRF-5 activity [47]. Thus, it can be assumed that EBV-stimulated CLL lymphocytes increase CD86 expression, but are unable to produce interferon due to impaired signaling via TLR.

In the control group, a significant increase in the percentage of CD19+/CD86+ lymphocytes was noted after 72 h of stimulation with EBV peptides, but not after adding PMA and ionomycin to the culture. Therefore, EBV antigens act on the CTLA-4:CD86 transmission pathway by multidirectional induction of the expression of these molecules both in vivo in CLL patients susceptible to EBV infection/reactivation and in vitro, increasing the percentage of CTLA-4+ and CD86+ B lymphocytes among human cells with no detectable amounts of EBV genetic material in PBMCs.

The conducted research allows for a thesis that the presence of EBV DNA is associated with an increased expression of molecules that inhibit signaling. Since in EBV(−) patients the absolute numbers and the percentages of CTLA-4 and CD86-positive lymphocytes were significantly lower, it seems that this virus is particularly responsible for the immunosuppression of the cellular response in CLL.

In our study, EBV DNA was measured at baseline only and thus should be deemed observational. Hence, we are not able to define how EBV status changes with time in CLL. Since as many as 90% of adults suffered from EBV infection, we believe that the detectable EBV DNA in most CLL patients suggests they have a reactivation of EBV infection rather than a new infection. In our study, patients with detectable EBV DNA had increased titers of reactivation markers (anti-EA and anti-VCA antibodies). On the other hand, all patients without detectable EBV DNA had antibodies against EBV antigens, which indicates previous EBV infection.

## 5. Limitations of the Study

This is a preliminary study; hence, the group was limited. We aimed to assess the significance of such research. Unfortunately, as our data collection of a much larger cohort has been strongly disrupted by the outburst of the COVID-19 pandemic, we decided to publish the preliminary results. Moreover, our study was not feasible to analyze clinical outcomes in patient subgroups that received different treatments due to the limited sample size. This limitation, however, did not affect the analysis of the time to treatment initiation, and findings regarding this end point were consistent with the time to lymphocyte doubling.

## 6. Conclusions

In conclusion, our studies have shown that anergy, which is expressed by inhibition through the interaction of CTLA-4 and CD86, is an important mechanism leading to the inhibition of the antitumor response. The disorders described above are closely related to the presence of EBV genetic material in leukemia cells and to the production of antibodies against antigens of this virus, which indicate their participation in the impaired cellular response in the course of CLL. Recent studies have shown that CLL patients do not constitute a homogeneous population. The question of which patients the disease will have a more rapid and more aggressive course remains unanswered. Developing parameters that can reliably predict the clinical course of CLL is a key issue. The presence of EBV DNA in CLL leukemia cells in a substantial proportion of patients in our research, having a significant impact on the biology and clinical course of the disease, suggests that EBV may become a new independent prognostic factor allowing the identification of patients with either poor or good prognosis. The obtained results so far have been very encouraging and justify further exploration of EBV infection in patients with CLL, in search of new treatment strategies of this disease incorporating antiviral treatment.

## Figures and Tables

**Figure 1 cancers-14-00672-f001:**
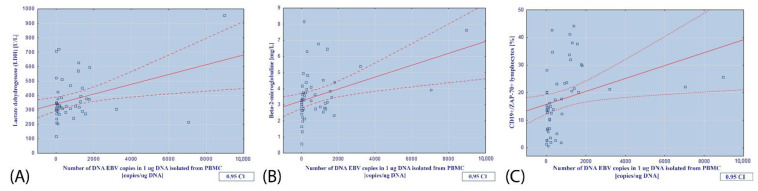
Relationship between the number of EBV DNA copies in 1 µg of DNA isolated from PBMCs and (**A**) the concentration of lactate dehydrogenase (U/L); (**B**) the concentration of beta-2-microglobulin (mg/L); (**C**) the percentage of CD19+/ZAP-70+ lymphocytes.

**Figure 2 cancers-14-00672-f002:**
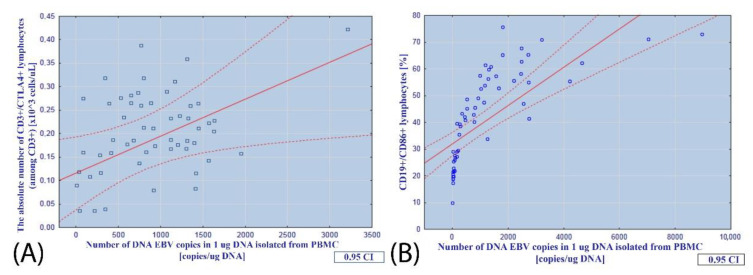
Relationship between the number of EBV DNA copies in 1 µg of DNA isolated from PBMCs and (**A**) the absolute number of CD3+/CTLA-4+ lymphocytes; (**B**) the percentage of CD19+/CD86+ lymphocytes in EBV(+) patients.

**Figure 3 cancers-14-00672-f003:**
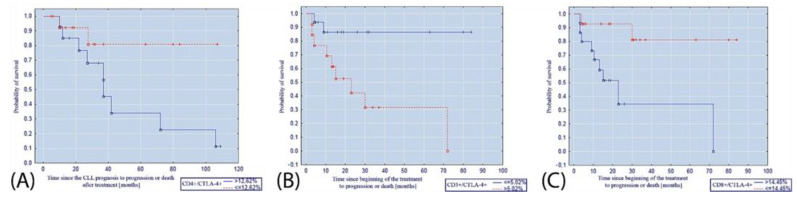
Kaplan–Meier curve illustrating the probability of progression-free survival (**A**) after diagnosis of CLL depending on the percentage of CD4+/CTLA-4+ lymphocytes; (**B**) after the initiation of CLL treatment depending on the percentage of CD8+/CTLA-4+ lymphocytes; (**C**) after the initiation of CLL treatment depending on the percentage of CD3+/CTLA-4+ lymphocytes.

**Figure 4 cancers-14-00672-f004:**
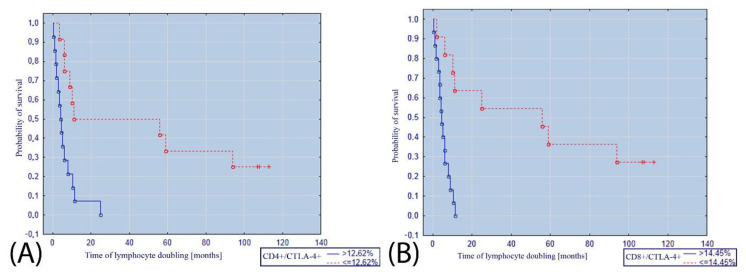
(**A**) Kaplan–Meier curve illustrating the probability of survival without doubling of the lymphocytosis depending on the percentage of (**A**) CD4+/CTLA-4+ lymphocytes; (**B**) CD8+/CTLA-4+ lymphocytes.

**Table 1 cancers-14-00672-t001:** Baseline characteristics of patients with CLL and control subjects.

	EBV(+) Patients with CLL	EBV(−) Patients with CLL	Controls	*p*
N	59	51	20	
AGE, YEARS, MEAN (SD)	63.81 (9.37)	63.63 (10.76)	64.50 (7.15)	0.943
WHITE BLOOD CELL COUNT G/L, MEDIAN [IQR]	28.30 [20.47, 48.92]	30.40 [19.85, 46.72]	7.02 [6.12, 7.85]	<0.001 ^a^
LYMPHOCYTE COUNT G/L, MEDIAN [IQR]	22.44 [15.40, 43.01]	22.58 [14.93, 40.55]	2.62 [2.19, 3.05]	<0.001 ^a^
HEMOGLOBIN, G/DL, MEAN (SD)	12.09 (1.42)	14.12 (1.27)	14.29 (1.19)	<0.001 ^b^
PLATELET COUNT, G/L, MEAN (SD)	150.02 (58.07)	180.51 (46.98)	279.00 (57.05)	<0.001 ^c^
BETA-2 MICROGLOBULIN, MG/DL, MEDIAN [IQR]	3.47 [2.78, 4.23]	2.37 [1.98, 2.87]	1.68 [1.29, 1.91]	<0.001 ^c^
LACTATE DEHYDROGENASE, U/L MEDIAN [IQR]	327.00 [290.50, 378.00]	254.00 [214.50, 334.00]	155.50 [137.00, 178.25]	<0.001 ^c^
RAI STAGE (%)				0.06
0	24 (40.7)	24 (47.1)	-	
I	11 (18.6)	15 (29.4)	-	
II	16 (27.1)	12 (23.5)	-	
III	2 (3.4)	0 (0.0)	-	
IV	6 (10.2)	0 (0.0)	-	
BINET CLASSIFICATION (%)				0.018
A	24 (40.7)	24 (47.1)	-	
B	27 (45.8)	27 (52.9)	-	
C	8 (13.6)	0 (0.0)	-	
CD19+ ZAP70+ CELLS > 20%, N (%)	27 (45.8)	11 (21.6)	-	0.009
CD19+ CD38+ CELLS > 30%, N (%)	34 (57.6)	6 (11.8)	-	<0.001
CD19+ ZAP70+ CELLS, % MEDIAN [IQR]	17.96 [11.09, 29.04]	5.80 [3.22, 18.19]	-	<0.001
CD19+ CD38+ CELLS, % MEDIAN [IQR]	33.53 [12.02, 59.89]	1.64 [0.78, 6.27]	-	<0.001

Notes: *p*-values are for comparisons using ANOVA (normally distributed variables) or the Kruskal-Wallis test. Frequencies were compared with Fisher’s exact test. When *p* < 0.05, post hoc comparisons were performed with Dunn’s test (*p*-values adjusted with the Benjamini–Hochberg method). ^a^
*p* < 0.05 on post hoc comparisons: controls vs patients with CLL both EBV(+) and EBV(−). ^b^
*p* < 0.05 on post hoc comparisons: EBV(+) patients with CLL vs EBV(−) patients with CLL and controls. ^c^
*p* < 0.05 on post hoc comparisons: EBV(+) patients with CLL vs. EBV(−) patients with CLL vs. controls. Abbreviations: CLL, chronic lymphocytic leukemia; EBV, Epstein-Barr virus; IQR, interquartile range; SD, standard deviation.

**Table 2 cancers-14-00672-t002:** CD86 expression in controls, EBV(−) patients with CLL, and EBV(+) patients with CLL.

Variable	Group	Median	Minimum	Maximum	*p*
LYMPHOCYTES CD19+/CD86+ [%]	EBV(+)	35.37	4.43	75.6	<0.0001 0.0002
	EBV(−)	16.73	1.5	46.99	
	Control	11.32	2.98	18.19	
ABSOLUTE NUMBER OF LYMPHOCYTES CD19+/CD86+ [X10^3^ CELLS/UL]	EBV(+)	6.4495	0.4433	32.656	0.0354 <0.0001 0.0256
	EBV(−)	2.5725	0.2094	22.0708	
	Control	0.0292	0.0108	0.0403	
CD86 ANTIGEN EXPRESSION ON LYMPHOCYTES CD19+ [MFI]	EBV(+)	21.83	14.18	131.24	0.0256 0.0067
	EBV(−)	18.655	14.24	29.44	
	Control	15.495	13.24	18.97	

**Table 3 cancers-14-00672-t003:** CTLA-4 expression in EBV(+) patients with CLL, EBV(−) patients with CLL and controls.

Variable	Group	Median	Minimum	Maximum	*p*
LYMPHOCYTES CD3+/CTLA-4+ (AMONG CD3+) [%]	EBV(+)	5.41	2.13	14.36	0.0395
	EBV(−)	4.715	1.95	8.06	
	Control	2.975	1.34	5.8	
ABSOLUTE NUMBER OF LYMPHOCYTES CD3+/CTLA-4+ (AMONG CD3+) [X10^3^ CELLS/UL]	EBV(+)	0.1433	0.0356	0.423	0.0078 0.0314
	EBV(−)	0.1515	0.0407	0.289	
	Control	0.0542	0.0129	0.0787	
LYMPHOCYTES CD4+/CTLA-4+ (AMONG CD4+) [%]	EBV(+)	14.755	9.56	31.14	0.0012 0.0001 0.0091
	EBV(−)	8.435	5.96	14.25	
	Control	2.85	2.17	3.19	
ABSOLUTE NUMBER OF LYMPHOCYTES CD4+/CTLA-4+ (AMONG CD4+) [X10^3^ CELLS/UL]	EBV(+)	0.1705	0.0913	0.908	0.0001 0.0150
	EBV(−)	0.1367	0.0761	0.2681	
	Control	0.0303	0.0147	0.0406	
LYMPHOCYTES CD8+/CTLA-4+ (AMONG CD8+) [%]	EBV(+)	16.855	11.38	26.69	0.0001 0.0001 0.0011
	EBV(−)	8.785	5.91	13.75	
	Control	2.835	1.58	3.89	
ABSOLUTE NUMBER OF LYMPHOCYTES CD8+/CTLA-4+ (AMONG CD8+) [X10^3^ CELLS/UL]	EBV(+)	0.1903	0.0322	1.487	0.0002 0.0420
	EBV(−)	0.1166	0.0295	0.2913	
	Control	0.0155	0.0101	0.0378	
EXPRESSION OF CTLA-4 ON CD8+ [MFI]	EBV(+)	28.185	26.22	34.52	0.0434
	EBV(−)	28.38	25.43	29.52	
	Control	22.895	17.57	37.65	
ABSOLUTE NUMBER OF LYMPHOCYTES CD19+/ CTLA-4+(AMONG CD19+) [X10^3^ CELLS/UL]	EBV(+)	1.1267	0.1395	8.676	0.0005 0.0019
	EBV(−)	0.9997	0.148	6.3925	
	Control	0.0056	0.0034	0.0203	

*p*-values are for comparisons with the Kruskal–Wallis test. When *p* < 0.05, post hoc comparisons were performed with the Dunn test; *p* < 0.05 on post hoc comparisons: EBV(+) patients with CLL vs EBV(−) patients with CLL vs controls; *p* < 0.05 EBV(+) patients with CLL vs EBV(+) patients with CLL and controls. CLL, chronic lymphocytic leukemia; EBV, Epstein-Barr virus; IQR, interquartile range.

## Data Availability

Due to privacy and ethical concerns, the data that support the findings of this study are available on request from the First Author, (E.G.).

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
