# Peer review of "Expression of CTLA-4 and CD86 Antigens and Epstein-Barr Virus Reactivation in Chronic Lymphocytic Leukemia—Any Link with Known Prognostic Factors?"

_cancers, 2022, doi:10.3390/cancers14030672_

Round 1

Reviewer 1 Report

The authors investigated the role of EBV in patients affected with chronic lymphocitic leukemia. Authors assessed, via PCR, EBV DNA in the blood of patients, the expression of CTLA-4 and CD86 on lymphocytes and on serum (via ELISA). Patients with CLL and EBV DNA +ve do bear a worst prognosis, potentially related to the upregulation of CTLA4 expression induced by EBV.

This is a well drafted manuscript, whose methodology seems robust and scientifically sound.

I would suggest native speaker revision to improve the quality of wirtten english.

Author Response

Dear Reviewer,
We sincerely thank you for time and effort that you dedicated to review our work and for its positive
review. It was corrected by an English native speaker as suggested.
With best regards,
Michał Mielnik

Reviewer 2 Report

The relationship between the presence of EBV DNA in CLL patients with adverse prognostic factors and poor outcome is interesting, but the sample  of patients and controls ( EBV - CLL patients) is too small for supporting the conclusions of the Authors. Moreover, no data are reported regarding the possible impact of different treatments on this high risk group of patients. Finally, the suggestion of a therapeutic strategy including chemoimmunotherapy and antiviral treatment is not supported by any significant specific experience.

Author Response

Dear Reviewer,
Thank you for a thorough review and distinguished questions. We really appreciate the time and effort
that you dedicated to providing your valuable feedback on our manuscript. We have been able to
incorporate the suggested changes into our work. Here is a point-by-point response to your
comments:
The relationship between the presence of EBV DNA in CLL patients with adverse prognostic factors
and poor outcome is interesting, but the sample of patients and controls ( EBV - CLL patients) is too
small for supporting the conclusions of the Authors.
This is a preliminary study, hence the group was small. We aimed to assess the significance of such
study. At the moment we are in the process of collecting a cohort of 1000 patients and we are
planning to check the expression of PD-1, CTLA-4 and other immune checkpoints for the presence of
EBV reactivation markers. Unfortunately our data collection has been strongly disrupted by the
outburst of the COVID-19 pandemic, so we decided to publish a preliminary study.
Moreover, no data are reported regarding the possible impact of different treatments on this high risk
group of patients.
Regarding the treatment - we studied untreated patients, at the time of diagnosis, the group was too
small to compare the amount of EBV DNA influenced by different therapeutic strategies - this is a topic
for a separate work and we will consider it with the above-mentioned larger group.
Finally, the suggestion of a therapeutic strategy including chemoimmunotherapy and antiviral
treatment is not supported by any significant specific experience.
We have added the “limitations of the study” section. The conclusions section was also redesigned to
incorporate your remarks. The article was also corrected by an English native speaker as suggested.
We sincerely hope you would deem the corrected manuscript suitable for publication.
With best regards,
Michał Mielnik

Reviewer 3 Report

The title is too long.

Please check for typos.

Why did you use Benjamini-Hochberg adjustment for some tests and Holm for others? Please explain.

Lines 216-217: Dichotomizing using the median values is not appropriate in most cases. Did you try other approaches?

Please note the version of R that you used.

Please review your tables. They look patchy.

The figures are hard to read. Moreover, why did you fit a linear model to the points in Figure 2?

Figure 3: I generally prefer to also have a table mentioning how many patients are left at certain timepoints.

Overall, the cohort size is rather small and the approach is not stringent enough to warrant reliable results.

Author Response

Dear Reviewer,
Thank you for a thorough review and distinguished questions. We really appreciate the time and effort
that you dedicated to providing your valuable feedback on our manuscript. We have been able to
incorporate the suggested changes into our work. Here is a point-by-point response to your
comments:
The title is too long.
The title was updated as suggested.
Please check for typos.
The article was corrected by an English native speaker to eliminate spelling mistakes.
Why did you use Benjamini-Hochberg adjustment for some tests and Holm for others? Please explain.
Please accept our apology. Mistakenly, we included the description of the method from our other
manuscript. In this study, it was not necessary to make a correction for p-values in multiple hypothesis
testing for log rank tests. The sentence "The p-values for log-rank tests were adjusted with Holm’s
correction." was deleted from 2.8. Statistical analysis paragraph
Lines 216-217: Dichotomizing using the median values is not appropriate in most cases. Did you try
other approaches?
Thank you very much for this important remark. From the clinical and diagnostic point of view, the
most important reference for us was the division of patients into EBV(+) and EBV(-) groups. The
reported median values coincided with the presence or absence of detectable EBV DNA. The
classification of CD86 and CTLA-4 expression as high or low was based on medians specific for the
subgroups of EBV(+) and EBV(−) patients.
Please note the version of R that you used.
Please forgive me for not clarifying R issue before. The R software version 3.0.2 (https://www.rproject.org) was used to complete all calculations.
Please review your tables. They look patchy.
The tables have been corrected for uniformity.
The figures are hard to read. Moreover, why did you fit a linear model to the points in Figure 2?
The quality of the figures was visibly decreased by the MDPI editorial system. I will contact the Editor
in order to resolve this issue.
Figure 3: I generally prefer to also have a table mentioning how many patients are left at certain
timepoints.
You have raised an important point here. However, we believe that another table would result in
overburden of data in that manuscript, hence we would prefer to leave figure 3 unchanged.
Overall, the cohort size is rather small and the approach is not stringent enough to warrant reliable
results.
This is a preliminary study, hence the group was small. We aimed to assess the significance of such
study. At the moment we are in the process of collecting a cohort of 1000 patients and we are
planning to check the expression of PD-1, CTLA-4 and other immune checkpoints for the presence of
EBV reactivation markers. Unfortunately our data collection has been strongly disrupted by the
outburst of the COVID-19 pandemic, so we decided to publish a preliminary study. We have added the
“limitations of the study” section.
We sincerely hope you would deem the corrected manuscript suitable for publication.
With best regards,
Michał Mielnik

Round 2

Reviewer 2 Report

The manuscript has been sufficiently improved to warrant publication. 

Reviewer 3 Report

Most comments have not been addressed appropriately.

I still consider that the manuscript should be rejected.